# Evaluation of Ti–Mn Alloys for Additive Manufacturing Using High-Throughput Experimental Assays and Gaussian Process Regression

**DOI:** 10.3390/ma13204641

**Published:** 2020-10-17

**Authors:** Xinyi Gong, Yuksel C. Yabansu, Peter C. Collins, Surya R. Kalidindi

**Affiliations:** 1School of Materials Science and Engineering, Georgia Institute of Technology, Atlanta, GA 30332-0245, USA; xinyigong@outlook.com; 2George W. Woodruff School of Mechanical Engineering, Georgia Institute of Technology, Atlanta, GA 30332-0405, USA; canyabansu@gmail.com; 3Department of Materials Science and Engineering, Iowa State University, Ames, IA 50011, USA; pcollins@iastate.edu

**Keywords:** high-throughput experimentation, additive manufacturing, Ti–Mn alloys, spherical indentation, statistical analysis, Gaussian process regression

## Abstract

Compositionally graded cylinders of Ti–Mn alloys were produced using the Laser Engineered Net Shaping (LENS™) technique, with Mn content varying from 0 to 12 wt.% along the cylinder axis. The cylinders were subjected to different post-build heat treatments to produce a large sample library of α–β microstructures. The microstructures in the sample library were studied using back-scattered electron (BSE) imaging in a scanning electron microscope (SEM), and their mechanical properties were evaluated using spherical indentation stress–strain protocols. These protocols revealed that the microstructures exhibited features with averaged chord lengths in the range of 0.17–1.78 μm, and beta content in the range of 20–83 vol.%. The estimated values of the Young’s moduli and tensile yield strengths from spherical indentation were found to vary in the ranges of 97–130 GPa and 828–1864 MPa, respectively. The combined use of the LENS technique along with the spherical indentation protocols was found to facilitate the rapid exploration of material and process spaces. Analyses of the correlations between the process conditions, several key microstructural features, and the measured material properties were performed via Gaussian process regression (GPR). These data-driven statistical models provided valuable insights into the underlying correlations between these variables.

## 1. Introduction

Modern metal additive manufacturing (AM) processes provide greatly expanded opportunities for producing engineered components possessing intricate geometries, novel material chemistries and internal structures. Furthermore, it is possible to tailor the material internal structures (hereafter simply referred to as microstructures) at different locations in the component both during the actual AM process and in subsequent (post-build) heat treatments in the effort to optimize its overall in-service functional performance [1,2,3,4,5,6,7]. Over the years, AM processes have been applied successfully in metal products with demonstrated benefits in net shaping, component repair, intricate geometry prototyping and component customization [8,9,10,11,12,13,14,15]. Although both experiments and physics-based multiscale material simulations have the potential to offer the data needed to gain insights into the correlations between the processing conditions and microstructure as well as the associated properties, our focus in this work was confined to experiments. This is mainly because the multiscale material models for AM metal alloys are still largely under development [16,17,18,19]. The AM process usually consists of multiple steps, including substrate treatment, powder delivery, energy delivery, nozzle movement and post-build heat treatments [20,21,22,23,24]. Each of these process steps involves the selection of multiple parameters that could significantly influence the local thermal history and, thereby, the microstructure and associated properties. The central challenges encountered in the experimental exploration of the influence of AM processing conditions and material microstructure come from two main gaps in current capabilities in the field. First, there is a lack of validated high-throughput experimental assays that are cost-effective and require only small amounts of material in different conditions (e.g., a range of chemical compositions and process histories). Second, there is also a lack of established approaches capable of building reliable data-driven process–structure–property (PSP) linkages from limited data (i.e., a relatively small number of data points). This is particularly important for AM metal alloy development using experimental assays, because it is unlikely that one can accumulate the very large datasets needed by conventional machine learning approaches such as neural networks [25,26,27,28].

A number of different experimental protocols have been explored in recent literature for the rapid formulation of PSP linkages from experiments in metal AM [29,30,31,32]. The central challenges come from the need to prototype a large library of samples spanning the large ranges of chemistries and process histories relevant to metal AM, and subsequently characterizing their microstructures and mechanical properties. In this respect, it should be noted that AM inherently offers many advantages in prototyping libraries of samples with small volumes. Previous studies [32,33,34,35] have demonstrated the feasibility of manufacturing compositionally and functionally graded materials with microstructural gradients using a selective laser-melting technique. Similarly, Joseph et al. [36] demonstrated the feasibility of exploring the vast compositional space of high-entropy alloys (HEAs) using the direct laser-fabrication technique. Beyond the prototyping of the sample libraries, one also needs to address the challenges in the high-throughput characterization of the samples. It is important to note that the characterization should include both details of the material microstructures and their mechanical properties in order to meet our target of extracting PSP linkages that can accelerate material innovation for AM. In recent work, Saltzbrenner et al. [31] have demonstrated the viability of prototyping miniaturized tensile test specimens and conducting high-throughput tests in automated protocols. Although this approach has tremendous potential, in practice, it is often challenging to extract reliable and consistently reproducible mechanical properties because of the large heterogeneity exhibited by the AM samples. Since tensile testing requires a statistically homogeneous material condition in the entire gauge section for the successful evaluation of mechanical properties, any significant variation in the local thermal histories at different locations of the gauge section of the tensile test specimen can lead to a large variance in the values of the measured mechanical properties from such measurements. For AM samples, there is a critical need to explore other characterization methods that can evaluate mechanical properties in small material volumes without the need to make standardized tensile test samples.

In recent work [37], it was demonstrated that it is possible to prototype compositionally graded AM samples and characterize their mechanical properties using the stress–strain analysis protocols based on spherical indentation. This strategy appears to exhibit tremendous potential for the high-throughput extraction of the relationships between the processing conditions, microstructural features and properties [38,39]. Although this strategy has produced reliable data points, the size of the dataset (i.e., the number of data points obtained) is still rather small for the extraction of PSP linkages using emergent machine learning techniques. In this paper, we demonstrate novel workflows that extend significantly the previously demonstrated assays in multiple research directions: (i) the prototyping of a much larger library of AM Ti–Mn alloys employing intentionally induced compositional gradients coupled with different post-build heat treatments, and (ii) the use of data-driven model-building strategies such as Gaussian process regression (GPR) [40,41,42,43,44,45,46] for extracting practically useful correlations from experimental datasets. GPR offers many potential advantages compared to other machine learning approaches, including the ability to utilize smaller data sets (i.e., smaller numbers of data points) [42,44], rigorous treatment of uncertainty [47,48] and dynamic selection of new experiments that maximize the expected information gain [49,50,51]. This work explores and demonstrates a framework for high-throughput experimental assays to facilitate the efficient exploration of the AM process space as well as statistical analyses of the accumulated data using GPR approaches.

## 2. Methods

### 2.1. Experiments

Laser Engineered Net Shaping (known as LENS) [52,53,54] is the prototypical powder-blown direct energy deposition technique used for additive manufacturing. It often incorporates computer-controlled lasers as power sources and produces near-net shapes with sufficiently accurate dimensions as the final product [55] to eliminate the need for rough machining, making it popular in industry [52,53,56,57]. Due to characteristics such as its great reliability [53,54,58] and the low porosity [59,60,61] of the final products, LENS is widely employed in the customization and repair of intricate mechanical parts, including turbine blades [54,62,63,64,65,66]. The ability to control, independently, the powder flow from separate powder feeders in LENS allows for creating chemical gradients in the AM components [67,68,69,70,71,72]. This is of tremendous interest for the present study, which aims to prototype a large library of material samples of small volumes covering a range of alloy compositions and post-build heat treatments. 

The binary Ti–xMn (x ranges from 0 to ~15 wt.% Mn) system was selected for this study. Titanium–manganese alloys are of great interest because of their numerous applications in aerospace, hydrogen storage and biomedical industries [73,74,75]. This range of manganese content in the alloy introduces a typical eutectoid β-stabilized system [73,76,77] that is notoriously susceptible to the segregation defect during solidification, known as “β-fleck” [78,79,80,81], and thus not suitable for traditional ways of developing cast/wrought titanium alloys. AM has the potential to eliminate β-fleck by taking advantage of the high thermal gradients and small molten pools, thereby reducing liquid-phase separation. By eliminating β-fleck, it may be possible to subsequently increase the strength through post-build aging heat treatments. An Optomec 750 LENS system was utilized to produce samples in this work. Elemental powders of Ti and Mn were introduced into the molten pool using two independently and automatically controlled powder feeders, one containing pure Ti and the other containing a mixture of elemental Ti and elemental Mn with a composition of 15 wt.% Mn. These powders, after leaving their powder feeders at preset feed rates, were mixed and focused into the molten pool by a multi-nozzle system. A Nd:YAG laser system producing near-infrared radiation with a wavelength of 1064 nm was focused coincident to the focal point of the powder, generating a local molten pool where melting and mixing occurred. The motion of the build plate was then controlled so that thin layers of controlled composition were deposited with predetermined width and thickness. The laser power at the molten pool was 410 W, and the nominal flow rate of the powders was ~2.6 g/min. The substrate travel speed (equivalent to the laser scan speed, but with a different reference frame) was 10 inch/min (the nonstandard units of inches and minutes are used when describing build parameters, as these are the standard units of the Optomec control system itself), and the hatch widths and layer thicknesses were 0.018 inch and 0.010 inch, respectively. The oxygen content in the glove box was maintained below 10 parts per million, with the balance being primarily argon gas.

Cylindrical samples with compositional gradients along their length were produced (see Figure 1a). Planning for the potential loss of volume of material due to cutting/machining/sample preparation (e.g., through the curfs of cuts), a small number of layers were programed to have the same composition at the beginning and end of the depositions. As a result, the samples produced for this study showed Mn content ranging from 0 to ~12 wt.% along the length. Three long strips (see the strip dimensions in Figure 1b) were sectioned out of the cylindrical sample and were subjected to different aging treatments. The aging treatments selected for the study included three different temperatures (500, 600 and 700 °C; see Figure 1c), while the aging time was kept the same, at four hours. The post-build aging treatment is expected to release residual stresses (these can be significant in the LENS technique due to the high power of the energy source, subsequent high temperature of the melt pool, fast cooling process and high build rate) as well as significantly alter the phase volume fractions and phase morphology, promoting possibilities of attaining improved properties.

After aging, all the sample strips were prepared for microscopy and spherical indentation stress–strain measurements using standard metallography protocols established previously for titanium alloys [82]. These included grinding (P240 and P1200 SiC papers), followed by polishing steps with decreasing abrasive particle sizes (9, 3 and 1 μm diamond suspensions), while making sure every step removed the surface deformation introduced by the previous step. A solution of 0.06 μm colloidal silica suspension with hydrogen peroxide in the ratio of 5 to 1 was employed in a final polishing step to produce the desired surfaces for microscopy and indentation. 

The main focus of this study is exploring high-throughput experimental assays for exploring large material spaces for AM. Five locations were selected longitudinally in each sample strip (see Figure 1b) for microstructure characterization and indentation tests. The transverse directions on the sample surface are not expected to exhibit any significant compositional gradients. Multiple indentation measurements and back-scattered electron (BSE) imaging were performed on a 5 × 5 grid at each of the five selected locations (illustrated in Figure 1d). Indentation tests were performed on an Agilent G200 (Santa Clara, CA, USA) with a continuous stiffness measurement (CSM) under a constant strain rate of 0.05/s and 800 nm indentation depth. The CSM was set at a 45 Hz oscillation with a 2 nm displacement amplitude [83]. A Tescan Mira XMH field emission SEM (Warrendale, PA, USA) with a 20 kV accelerating voltage was used to capture back-scattered electron (BSE) images. Energy dispersive spectroscopy (EDS) was performed at the five locations shown in Figure 1b to measure the Mn content. At each location, five EDS measurements randomly distributed within the 5 × 5 grid (established in Figure 1d) were performed. Each measurement was carried out by first mapping the element concentration distribution of a 50 μm × 50 μm area and then calculating the average element composition according to the map. A Hitachi SU8230 SEM (Tokyo, Japan) equipped with Oxford EDAX and Aztec analysis software was used for EDS analysis. Beam calibrations with a 100% copper plate were used for EDS quantification. The accelerating voltage was kept at 20 kV and beam intensity at 20 µA for these measurements.

### 2.2. Microstructure Analysis and Quantification

The two-phase BSE images were segmented with Otsu’s method [84,85]. Otsu’s method separates the intensity distribution of an image into two classes by using a threshold. The threshold value is determined to maximize the interclass variance (or minimize the intraclass variance). Otsu’s thresholding was performed using the “graythresh” function of the numerical computing software MATLAB [86]. The segmented (binary) images were used to compute the volume fraction of the β phase. Additionally, averaged chord lengths (CL) [87,88] were computed to quantify the length scales of the α and β phase regions in the microstructure. The procedures used to identify the chords are based on pixelized representations of the images and have been described in prior work [87,89]. A chord is defined as a line segment (measured as the number of pixels) that completely lies inside a distinct material phase, whose extension in any direction by even one pixel encounters pixels of a different material phase. 

### 2.3. Mechanical Characterization

The spherical indentation stress–strain protocols [90,91,92] employed in this study are built largely on Hertz’s theory [93,94] for elastic frictionless contact between two isotropic bodies with parabolic surfaces (see Figure 2a). The relevant relationships are summarized below: (1)$$P=\frac{4}{3}E_{eff}R_{eff}{}^{1/2}h_{e}{}^{3/2}$$
(2)$$a=\sqrt{R_{eff}h_{e}}=\frac{S}{2E_{eff}}$$
(3)$$\frac{1}{E_{eff}}=\frac{1-v_{i}^{2}}{E_{i}}+\frac{1-v_{s}^{2}}{E_{s}}$$
(4)$$\frac{1}{R_{eff}}=\frac{1}{R_{i}}+\frac{1}{R_{s}}$$
where P and $h_{e}$ denote the indentation load and elastic indentation displacement, $E_{eff}$ and $R_{eff}$ denote the effective modulus and the radius of the indenter-sample system, subscripts i and s correspond to the indenter and the sample, and the Young’s modulus and Poisson’s ratio are denoted as E and ν. In Equation (2), S (= $dP/dh_{e}$) denotes the elastic stiffness (also known as the harmonic stiffness in continuous stiffness measurement (CSM) protocols [83,95,96]). Building on these relationships, one can define the indentation stress, $\sigma_{ind}$, and the total indentation strain (includes the elastic and plastic strains), $\varepsilon_{ind}$, as
(5)$$\sigma_{ind}=\frac{P}{\pi a^{2}}$$
(6)$$\varepsilon_{ind}=\frac{4}{3\pi }\frac{h_{s}}{a}$$
where $h_{s}$ is the corrected sample displacement (subtracting the displacement in the indenter, $h_{i}$, from the total displacement, h) and is computed using
(7)$$h_{s}=h-\frac{3\left(1-v_{i}^{2}\right)P}{4E_{i}a}$$

The indentation stress and indentation strain defined in Equations (5) and (6) exhibit a linear relationship in purely elastic indentations, where the slope of the indentation stress–strain curve is referred to as the indentation modulus, $E_{ind}$. For an isotropic material, the indentation modulus and the Young’s modulus are related as
(8)$$E_{ind}=\frac{E_{s}}{\left(1-\nu_{s}^{2}\right)}$$

On the indentation stress–strain curve (see Figure 2b), a 0.2% offset indentation plastic strain is used to define the indentation yield strength, $Y_{ind}$. The indentation stress–strain curve between the 0.5% and 2% offset indentation plastic strains is fitted with a linear regression [82,97] to compute the indentation work hardening rate, $H_{ind}$. In prior work [82,91,98,99], Equations (5) and (6) were demonstrated to produce meaningful elastic–plastic indentation stress–strain curves that show an elastic–plastic regime following an initial elastic regime (see Figure 2b).

In the present study, spherical indentations on a 5 × 5 grid were performed with a uniform spacing of 100 μm (see Figure 1d). A diamond indenter tip with a nominal radius of 100 μm was used in all the tests reported in this work. Each indentation produced a contact area of about 150 μm^2^ (contact radius of roughly 7 μm) at indentation yield and hence reflected the effective response of the two-phase microstructures obtained in the sample library (micrographs presented later). The spacing between indentations was designed to be large enough to minimize the interference between neighboring indentations. However, it was also important to keep the spacing small enough so that the compositional variation between the indentation locations within each grid was very small. 

### 2.4. Gaussian Process Modeling

In this study, Gaussian process regression (GPR) was employed to establish quantitative correlations between various measured quantities of interest in the extracted dataset. GPR is a nonparametric machine learning method that employs joint probability distributions to the available training data (usually a small dataset) in order to make probabilistic predictions for new inputs. This is accomplished by treating the correlations as a Gaussian process (GP) defined by only its mean and variance. Let $t=\left\{t_{1},\;t_{2},\;\ldots ,t_{N}\right\}$ and $y=\left\{y_{1},\;y_{2},\;\ldots ,y_{N}\right\}$ denote the target and prediction, respectively, where N denotes the number of training points. Then, the relationship between the target t and prediction y can be written as
(9)t=y+ε
where ε is a column vector containing the residuals of N observations. A GP governing the joint distribution between the predictions can be written as
(10)$$y\left(x\right)\;\sim N\left(\mu \left(x\right),K\left(x,x^{\prime }\right)\right)$$
where x denotes a 1×D input vector, and μ(x) and $K\left(x,x^{\prime }\right)$ represent the mean and the covariance of the GP, respectively. In Equation (10), N() denotes a multivariate Gaussian distribution. 

The covariance of the GP is generally computed using a kernel function $k\left(x,x^{\prime }\right)$. In the present study, the %Mn and the post-heat treatment temperature are treated as the two (i.e., D=2) independent variables (i.e., inputs) for the model-building effort in this study. The outputs for the study will include a number of microstructure statistics as well as the measured mechanical properties. The automatic relevance determination squared exponential (ARDSE) [100,101,102] was selected as the kernel for computing the covariance matrices. This ARDSE kernel is mathematically expressed as
(11)$$k\left(x,x^{\prime }\right)=\sigma_{f}^{2}exp\left(-\frac{1}{2}\left(\frac{{\left(x_{T}-{x\prime }_{T}\right)}^{2}}{l_{T}^{2}}+\frac{{\left(x_{c}-{x\prime }_{c}\right)}^{2}}{l_{c}^{2}}\right)\right)+\sigma_{n}^{2}\delta_{xx^{\prime }}$$
where $\sigma_{f}$, $l_{T}$, $l_{c}$ and $\sigma_{n}$ are the hyperparameters that control the fidelity of the GP model, and the subscripts T and c refer to the two input variables (i.e., the post-heat treatment temperature and %Mn). The hyperparameters in the kernel provide more valuable information about the trends and relationships between the inputs and the outputs, especially when compared to conventional correlation techniques such as the Pearson correlation coefficient [103]. More specifically:
(1)$\sigma_{f}$ is called the output scaling factor and determines the variance of the output values. A higher value of $\sigma_{f}$ indicates that the values of the output are widely spread. The ratio of $\sigma_{f}$ to the output noise $\sigma_{n}$ (discussed later) determines the uncertainty of the predictions made from the GP model.(2)$l_{T}$ and $l_{c}$ are the interpolation length scale parameters associated with the two input variables and capture the sensitivity of the output variable to the changes in the respective input values. Lower length scale values exhibit shorter memory, leading to sharper fluctuations and more complex nonlinear mapping between the inputs and the output. In other words, lower values of the interpolation length parameter indicate a higher sensitivity of the output to the input value (for the selected input variable). Conversely, larger values of the interpolation length parameters indicate low levels of correlation between the output and the corresponding input variable. (3)$\sigma_{n}$ is called the output noise hyperparameter and captures the variance in the training data. For the present study, where the training data are obtained from experiments, this variance can arise from variations in the execution of the experimental assays themselves or variations in the application of the analysis protocols (e.g., image segmentation). $\sigma_{n}$ is assumed to be the same for the entire input domain (also called homoscedasticity [104]).

The hyperparameters in Equation (11) are generally optimized to produce the most reliable predictions for test data points. For this, one must formulate a conditional distribution of test points, $t^{*}$, given the evidence of training points, t. Let the train and test datapoints be represented by matrices X and $X^{*}$ of sizes N×D and $N^{*}\times D$, respectively, where $N^{*}$ reflects the number of test points. The overall covariance matrix can be partitioned as
(12)$$C=\left[\begin{array}{cc} K\left(X,X\right) & k^{*}\left(X,X^{*}\right)\\ k^{*}{}^{†}\left(X,X^{*}\right) & K^{*}\left(X,X\right) \end{array}\right]$$
where † represents the transpose. Each term of the covariance matrix in Equation (12) is computed using the kernel function from Equation (11). The predictive distributions for the test points, given the training points, can be expressed as [100,101].
(13)$$\begin{array}{c} \mu^{*}=k^{*†}K^{-1}t\\ \Sigma^{*}=K^{*}-k^{*†}K^{-1}k^{*} \end{array}$$
where $\mu^{*}$ and $\Sigma^{*}$ denote the prediction means and variances (i.e., uncertainty), respectively, for the test points. The central challenge in the computations described in Equation (13) comes from the need to perform an inverse on the N×N covariance matrix of the training points, which requires $O\left(N^{3}\right)$ computations. Once $K^{-1}$ is obtained, predictions for the test points can be realized through standard matrix multiplication/addition operations, which require only $O\left(N^{2}\right)$ computations [100,101]. Note, also, that in the applications explored in this work, the number of data points is quite small. Therefore, the one-time computational cost of the inversion operation in Equation (13) does not represent a major challenge for the present study.

## 3. Results

Figure 3a shows a typical BSE micrograph taken from a location approximately 14 mm away from the pure titanium end of the sample strip aged at 500 ℃. In this micrograph, the lamellated hcp (hexagonal closest packing) α-Ti and bcc (body-centered cubic) β-Ti are visible as the darker and brighter regions, respectively. The EDS measurements also show less than 1 wt.% of manganese for the darker phase and about 15 wt.% of manganese for the brighter phase, thereby identifying these regions as α and β titanium, respectively. A map sum spectrum was also taken, measuring the average manganese content at 5.8 wt.% for this scan. Similar measurements were carried out at each location identified in Figure 1b for each sample strip (i.e., each composition–post-heat treatment combination). The results are presented in Figure 3b. As expected, it is seen that the variation of the Mn content along the strip is highly consistent between the different strips. The manganese composition rises from the pure titanium end but peaks at about 20 mm and stays at about 12 wt.%. Note that this 12 wt.% Mn is a little lower than the target composition of 15 wt.%. The maximum manganese composition is present over a few millimeters at the end of the build, as programmed into the original motion control source code. The deviations between the obtained local composition and the targeted composition are attributed to variations in the local elemental powder being fed or, as is the case here, due to an intentional extension of a region with the maximum Mn concentration. Since our primary interest in the present study is the development of a framework for establishing the correlations between the processing parameters, microstructure statistics and the properties, we have not iterated with different starting powder mixtures to attain specific compositions in the produced samples. Instead, our focus will be on the protocols needed to acquire, efficiently, the material data needed for the targeted correlations. 

Multiple BSE micrographs were obtained corresponding to each combination of manganese composition and post-heat treatment temperature. The volume fractions estimated from the segmented images are shown in Figure 4. It is seen that the β volume fraction increased with Mn content and with the temperature of the post-build aging treatments. This is because post-build aging at a higher temperature pushes the microstructure to be close to its equilibrium state. Note that the high manganese locations subjected to the low 500 °C treatment (see the bottom left micrograph in Figure 5) produced a small-scale (10–100 nm) secondary α phase [33] in addition to the bigger (~2 μm) primary α laths. Such secondary α is expected, especially at these lower temperatures, and results when new nucleation events become more favorable and accelerate the rate of transformations.

For the computation of the averaged CLs, all the chords in the micrograph were collected at intervals of 2.5 degrees to avoid imaging orientation bias. The averaged value of all the collected chord lengths for each phase at each of the five sample locations identified is reported in Figure 6. The averaged CL of the dominant β phase decreased consistently with an increase in the Mn content. By contrast, the averaged chord length of the β phase increased with a higher manganese content, with the higher aging temperature promoting a more drastic change. 

Spherical indentation tests were performed on a grid of twenty-five sites for each of the five sample locations (see Figure 1b,d) for all three sample strips. Figure 7 summarizes the measured values of elastic moduli, indentation yield strengths and indentation hardening rates at each of the five locations on all three strips studied in this work. It is observed that the measured indentation moduli did not show significant variations between different locations and between different sample strips. On the other hand, a strong positive correlation was observed between the Mn content (which also correlated well with the beta volume fraction (see Figure 4)) and the indentation yield strength as well as the indentation hardening rate.

It is clearly seen that the various microstructural features (β volume fraction and the averaged CLs of the α and β regions) and the resulting mechanical properties are highly correlated with each other. In order to analyze the effects of the process conditions (i.e., the aging temperature and Mn content) on the microstructural features and the resulting mechanical properties, it is necessary to conduct a statistical analysis. Gaussian process regression (GPR) was employed in this work for this purpose. As mentioned before, the hyperparameters of the kernel function provide reliable insights into the sensitivities of the different inputs to the outputs of interest. 

A separate GP was built for each of the six outputs listed in Table 1, while using the post-build aging temperature and Mn content as features (i.e., independent variables). Traditionally, GP models are built to provide predictions for new inputs. However, in the present application, the size of the dataset is too small to formally establish a reliable predictive model with rigorous cross-validation. Therefore, it was decided to use the GP models to provide reliable insights into the sensitivities between the various measured quantities in this study. The interpolation length scale parameters established by these GP models and summarized in Table 1 are ideally suited for extracting such insights. As a specific example, it is seen that the interpolation length scale hyperparameter for the aging temperature in the GP model for the averaged CL for the α phase is very large, especially compared to the corresponding values obtained for the GP models for the other five outputs. This indicates a much lower sensitivity of the averaged CL of the α phase to the aging temperature. In fact, the averaged β-CL and the indentation hardening rate are found to exhibit the highest levels of sensitivity to the aging temperature. The table also indicates that all the microstructural parameters exhibited strong sensitivity to the Mn content, with the β volume fraction showing the highest sensitivity. 

A comparison of the output scaling factor $\sigma_{f}$, which controls the overall spread of the output values in the entire dataset with the output noise parameter $\sigma_{n}$, provides insight into the combined overall predictive capability of the GPR model. The ratio $\sigma_{f}/\sigma_{n}$ is referred to as the output-to-noise ratio and reflects the capability of the selected inputs in influencing the predicted output. For example, a very high value of $\sigma_{f}/\sigma_{n}$ obtained in a specific GPR model indicates that the selected inputs (i.e., the Mn content and the aging temperature) are able to reliably account for most of the observed variations in the selected output in the collected dataset. In other words, the GP models with high values of $\sigma_{f}/\sigma_{n}$ are indeed more mature and can be used reliably in making predictions for new inputs. In Table 1, it is seen that the GPR models for the elastic modulus and the β volume fraction show very high values of $\sigma_{f}/\sigma_{n}$, indicating that these models are able to account for almost all of the measured variations in these quantities in the data aggregated in this work. Similarly, a low value of $\sigma_{f}/\sigma_{n}$ might suggest a lack of adequate correlations between the selected inputs and the output. This could suggest that there is inherently more noise in the measured values of the selected output, the possible existence of as-yet-unidentified inputs influencing the output variable, or both. In Table 1, the lowest value of $\sigma_{f}/\sigma_{n}$ was obtained for the averaged CL for the β regions. In this study, we believe this is because of the inherent noise resulting from the protocols used to estimate this attribute from the micrographs (i.e., the segmentation and CL protocols). In other words, if one intends to establish more accurate correlations for the averaged CL for the β regions, it would be prudent to improve the protocols used to extract this value.

Table 1 also summarizes the mean absolute percentage error (MAPE) using a leave-one-out cross-validation strategy. This entails obtaining a model by setting aside one data point at a time in establishing the GPR model and subsequently testing the obtained model on the excluded point. The process is then systematically repeated for all available data points, and the MAPE is computed based on the obtained errors. It is seen from Table 1 that the GPR model for the elastic modulus exhibits the highest accuracy, while the GPR models for the averaged CL for the β regions exhibited the lowest accuracy. It is also seen that this is consistent with the $\sigma_{f}/\sigma_{n}$ values. 

## 4. Discussion

Compared to the conventional assays, the high-throughput assays employed in this work required significantly smaller material volumes. It should be noted that a total of 15 material conditions (obtained by combining three different post-build aging heat treatments with five different compositions) were produced and studied with relatively low overall effort and cost. In fact, the high-throughput (HT) assays described in this work exhibit tremendous potential for further scale-up, allowing the rapid evaluation of several hundreds of material conditions. In addition to requiring only small volumes of the material, the time and effort needed for the proposed HT assays are also significantly lower compared to those for the conventional assays. This is because the sample preparation steps only require standard metallography protocols that are needed anyway if the material microstructure is to be documented in such explorations. 

As demonstrated in previous studies [37,39,82,105], the averaged values from the multiple indentations summarized in Figure 7 provide reliable measures of the bulk properties measured in standardized tests. The estimated Young’s moduli did not show clearly identified trends between the different material conditions explored in this study, and fell in the range of 97–130 GPa. In general, one might expect a decrease in the Young’s modulus with an increase in the β volume fraction, as the β phase is expected to exhibit a lower elastic modulus compared to the α phase [106,107]. Although one might be tempted to infer such a trend from Figure 7, it is not clearly evident, as the noise in the measurements is of the same order as the overall variation among the tested samples. However, the overall range of the values estimated in this work is comparable to the ranges published in the literature [108,109] for similar compositions.

As seen in Figure 7, the indentation yield strengths and the indentation hardening rates in the early stages of the imposed plastic deformation increased systematically with the increase in Mn content. There is, as expected, a clear positive correlation between the indentation yield strength, the indentation hardening rates and the Mn content. Strengthening due to secondary phase, solid solution strengthening, and grain boundary strengthening are likely to contribute to the observed increase in the indentation strength with the increase in Mn content. Based on prior work [110], the indentation yield strengths can be converted to tensile yield strengths using a scaling factor of 2.0. Using this scaling factor, the tensile yield strengths for the material conditions studied are expected to range between 828 and 1864 MPa, which is among the highest ranges reported [108,111,112,113] for similar compositions. Interestingly, the highest values were obtained for the samples with the highest Mn content and the lowest post-build aging treatment. Indeed, the corresponding microstructures also showed a relatively high β volume fraction of about 65% and highly refined microstructures with averaged CLs of 0.17 and 0.32 µm in the α and β phases, respectively (see Figure 5). The refined length scales are also responsible for the high indentation hardening rates measured in our experiments, because of the presence of a much larger number of interfaces per unit volume of the material. The fact that our high-throughput protocols easily identified the viability of obtaining a very high yield strength in the Ti–Mn alloys along with the features identified in their microstructures clearly testifies to the unique benefits of our approach for the rapid exploration of large material spaces.

## 5. Conclusions

Novel high-throughput assays have been proposed and demonstrated to rapidly explore large material spaces reflecting the many combinatorial selections in material compositions and AM process parameters such as post-build aging treatments. More specifically, this study successfully conducted such an evaluation using Ti–Mn alloy systems processed by LENS, which allowed for the generation of samples with controlled composition gradients. Combining this strategy with spherical indentation stress–strain protocols allowed for a rapid exploration of the mechanical properties of the produced samples in small material volumes. Most importantly, this rapid exploration revealed that a Mn content of about 12% with a post-build heat treatment of 500 °C produced an unusually hard material with an expected tensile yield strength of 1864 MPa. The dataset generated in this study was analyzed rigorously using GPR models. The use of these statistical approaches revealed that the use of the Mn content and the post-build aging treatment as inputs does lead to reliable correlations with microstructure measures such as the β volume fraction and the averaged CLs of the α and β regions, as well as their mechanical properties such as the Young’s modulus, indentation yield strength and indentation hardening rate. These correlations revealed the relative sensitivities of the different outputs to the selected inputs as well as the high levels of inherent noise in the estimation of the averaged CLs of β regions. The GPR models built with the limited data obtained in this work showed reasonable accuracy in leave-one-out cross-validation. This study established the feasibility and value of employing GPR approaches in the rigorous statistical analyses of the datasets produced in the proposed high-throughput assays for material exploration.

## Figures and Tables

**Figure 1 materials-13-04641-f001:**
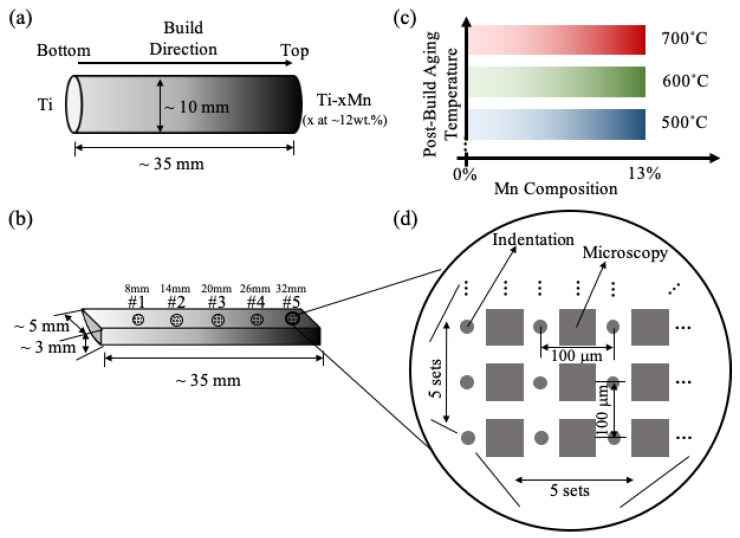
(**a**) Illustration of the layered Ti–Mn cylindrical sample manufactured by the Laser Engineered Net Shaping (LENS) process in this study. (**b**) Sample strip sectioned from (**a**) with compositional gradient along the length of the sample. Five locations were chosen longitudinally in each sample strip for characterization. They were 8, 14, 20, 26 and 32 mm away from the pure titanium end of the strip (labeled as #1–#5, respectively). (**c**) Three different sample strips were aged at three different temperatures (500, 600 and 700 °C, respectively) for four hours to produce the sample library used in this work. (**d**) A grid of indentation and microscopy characterization was performed at each location illustrated in (**b**). Each circle represents an indentation testing site, while the square represents the microscopy characterization site. Each measurement grid contained 5 by 5 indentation tests and the same number of microscopy characterizations. The test points in the grid were evenly spaced at 100 μm. Note all test sites shown in (**b**) are intentionally kept away from the thin end of the sample strips, making sure the sample has at least 2 mm thickness at the indentation test sites.

**Figure 2 materials-13-04641-f002:**
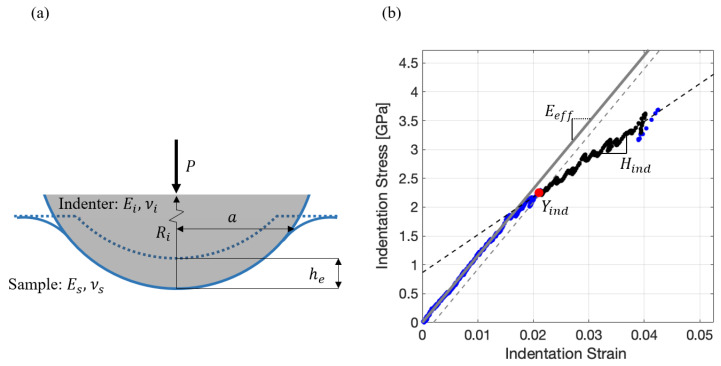
(**a**) Illustration of spherical indentation. (**b**) Indentation stress–strain curve acquired from Location #4 (see Figure 1b) of the strip heat treated at 700 °C. The slope illustrated in the elastic portion of the indentation stress–strain curve is the effective modulus, $E_{eff}$. The red dot represents the indentation yield strength $Y_{ind}$ corresponding to a 0.002 offset indentation plastic strain, while the black segment (from 0.005 to 0.02 in offset indentation plastic strain) represents the data used to estimate the indentation work hardening rate $H_{ind}$.

**Figure 3 materials-13-04641-f003:**
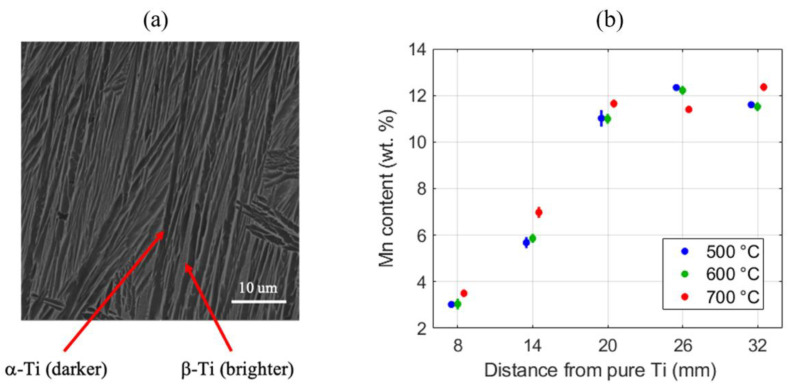
(**a**) Back-scattered electron (BSE)-SEM image for the sample strip aged at 500 ℃ for four hours and at the location where the Mn content was 5.8 wt.%. It depicts the dual-phase microstructure of the sample, where the darker phase is α-Ti and the brighter phase is β-Ti. (**b**) Means and standard deviations from the energy dispersive spectroscopy (EDS) measurements of the Mn content at the five locations for all three high-throughput (HT) sample strips produced for this study. For clarity, all 500 °C and 700 °C values are intentionally shifted slightly in the negative and positive x directions, respectively. All points in each group correspond to the same nominal distance indicated by the axis ticks.

**Figure 4 materials-13-04641-f004:**
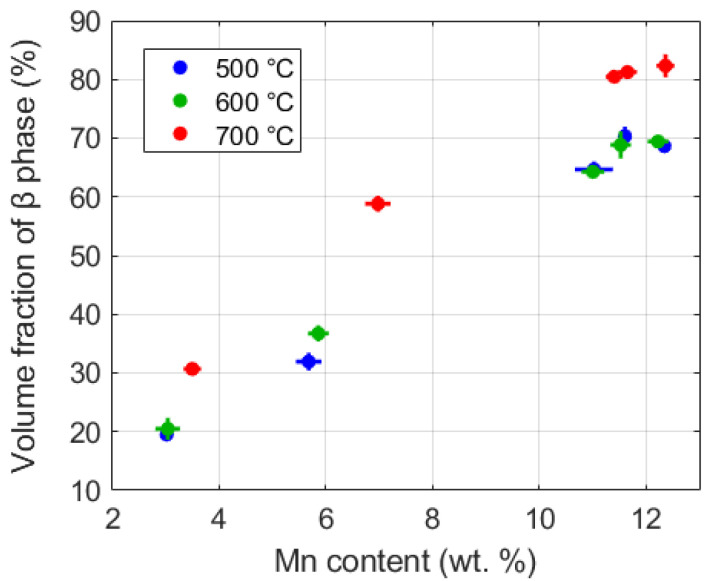
Means and standard deviations of the percentage volume fractions of the β phase obtained for the different Mn contents and post-build aging heat treatments.

**Figure 5 materials-13-04641-f005:**
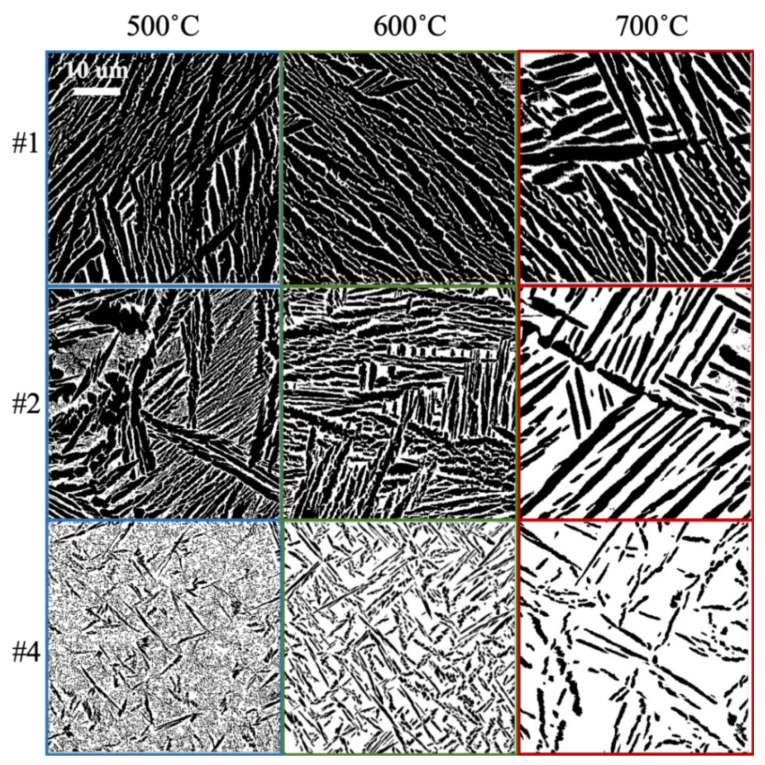
Segmented SEM-BSE images for the sample library produced and studied in this work. The left, middle and right columns correspond to aging heat treatments of 500, 600 and 700 °C, respectively. The rows correspond to different locations exhibiting different manganese compositions (see Figure 1b and Figure 3b). The black phase in these micrographs represents α-Ti, while the white phase represents β-Ti.

**Figure 6 materials-13-04641-f006:**
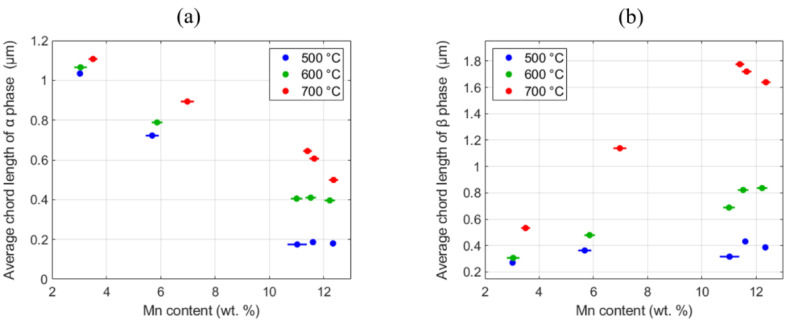
Averaged chord lengths (CLs) of (**a**) α phase and (**b**) β phase at the selected five locations for all three high-throughput (HT) sample strips studied in this work.

**Figure 7 materials-13-04641-f007:**
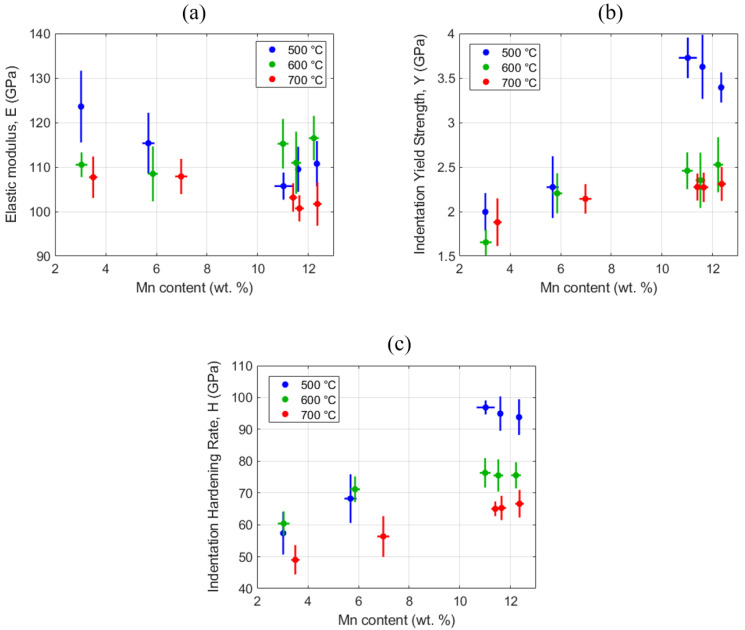
Mechanical properties estimated from the spherical indentation stress–strain protocols: (**a**) Young’s modulus, (**b**) indentation yield strength and (**c**) indentation initial hardening rate. The blue, green and red boxes correspond to the 500, 600 and 700 °C aged strips, respectively.

**Table 1 materials-13-04641-t001:** Gaussian process regression (GPR) interpolation length hyperparameters and the mean absolute percentage error (MAPE) for each of the six outputs selected for these models. CL denotes the averaged chord length, VF is the volume fraction, Y is the indentation yield strength, and E is the Young’s modulus.

GPR Results	CL α	CL β	VF-β	Y	H	E
$l_{T}$	667.17	141.85	263.23	199.12	143.38	211.78
$l_{c}$	10.95	10.48	9.39	16.83	14.78	12.49
$\sigma_{f}$	19.52	18.47	0.55	2.48	53.97	92.17
$\sigma_{n}$	0.93	1.66	0.01	0.17	2.64	1.39
$\sigma_{f}/\sigma_{n}$	20.88	11.13	45.59	14.50	20.41	66.10
*MAPE*	6.89	9.87	3.54	6.26	3.32	2.02

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
