# Peer review of "Evaluation of Ti–Mn Alloys for Additive Manufacturing Using High-Throughput Experimental Assays and Gaussian Process Regression"

_materials, 2020, doi:10.3390/ma13204641_

Round 1

Reviewer 1 Report

The manuscript under the title: “Evaluation of Ti-Mn Alloys for Additive Manufacturing using High Throughput Experimental Assays and Gaussian Process Regression”  is relevant for the Materials journal. The authors work on up-to-date topic connected with titanium materials. The article based on original experimental research. The organization of the article is appropriate. The abstract is very informative. Overall, the paper is well prepared, but there are needed small improvements, in order to be published:

- Please add the manufacturing details (year, city) of the equipments

- Please add the DOI links for all references.

Reviewer 2 Report

  In this paper the authors investigated methodology to evaluate the mechanical properties of metal materials using the spherical indentation protocols. They adopted compositionally-graded Mn-Ti alloy-cylinders as the targeted material. Further investigation is required to judge whether this approach is valid or not, although the approach is challenging to worth investigating as it may open a way to rapidly evaluate the mechanical properties of a material.

  A statistic approach with Gaussian process regression helped phenomenological understanding on the relationship between microstructural measures such as the volume fraction of beta phase and CLs and the mechanical properties such as Young’s modulus, indentation yield strength, and the hardening rate of indentation.

  Redundancy of the expressions is a drawback of this paper and revising them will make this paper more effective.

  The discussion is sufficiently supported by the experimental data. The reviewer does not have any comment on the contents. He recommends Accept for this paper after minor revision. It is advisable to reconsider the descriptions which may be written more concisely.

Reviewer 3 Report

Dear authors,

In general, this is quite an interesting work.
However, there are quite some detracting points that needs to be improved before this paper is on a publishable level, those are for instance:
1. The structure of the paper is rather mixed, for instance, there are some sentences/paragraphs that would fit better in the Introduction section located in the Materials and Method section (for example in line 100-104, about LENS, this more suited to be placed in the Introduction section).
2. Also about mixed structure, in the Result section, the authors often mixed the description of the result with some discussion (e.g. in line 309-312), you have written "Because the combinatorial efforts described here aim to 309 correlate the measured local composition and microstructure with the corresponding measured local properties, the variation between the targeted composition and what was actually obtained in our samples is not relevant.".
3. In Figure 3b, why does the points of 700°C slightly shifted a bit to the right from the nominal positions (i.e. 8,14,20,26,32mm) while the 500 is slightly moved to the left, only 600 °C is the more or less in nominal position? This also holds true for Figure 7a-c, why does when the alloy heat treated at 500 °C the composition is lower than when it is heat treated at 700°? Please provide some explanation and supporting references.
4. There is lack of information in the Introduction section, for instance on:
- The materials of interest in this study: Ti-Mn. Why do the authors decided to study this material? (i.e. in which applications this material is usually used for? is this alloy is heat treatable? etc. and please put references)
- Why this alloy is interested to be applied for AM?
- Why do the authors decided to study up only until 15wt.%, any special reason for this selection?

5. In line 113, can you give brief description of B-fleck here as it would help reader rather that they have to find/access the refrence journal?
6. When you said LENS is sufficiently accurate (in line 102), could you please quatify the accuracy (or perhaps compare it with other AM technique/production technique)?
7. In line 127, you menstioned in/min and then there's a 1 in the superscript, what does this means?
8. How many samples per conditions was built? It seems you only produced one sample for each heat treatment condition, could you please comment regarding the repeatibility ? And since this is a statistical study, why don't you make like three samples? Comment about repeatibility please.
9. It would be nice to add the positions in mm (Would be nice to be added in Figure 1b to help the reader) for each testing locations (i.e. position #1, #2, #3 ...etc) in Figure 1b to help the reader.
10. In Section 2.2., the authors mentioned segmentation is performed using Otsu's method, but using which software? And perhaps any special settings for image processing steps?
11. Why does the composition is not linear across the length axis of the sample, it seems the amount of Mn does not linearly increases across the length of the sample (from 0 to 32 mm).
12. Could you please comment on the properties of the as-made sample (no-heat treatment) in terms of Mn content, alpha and beta content also mechanical properties at different level of Mn contents as a baseline? This will be useful for the reader how the heat treatment affects the physical properties of the alloy.
13. It seems in this work you only produced 1 sample for each heat treatment condition (one for 500 °C, one for 600 °C and one for 700°C). Could you please comment on the repeatibility of the produced samples (in terms of microstructure and composition for example)? Can you quantify it? And since this is a statistical analysis, why don't you discusss the repeatibility and statistical analysis of three samples processed at same heat treatment route, for example?
14. In line 104, you mentioned that the maximum Mn composition produced in this alloy is around 3wt% lower than target (15 wt%)? Could you please provide reasons for this? Is it an experimental artifact, why is this occurence happens? Because 3wt.% below the target (15wt%) is quite significant, around 20%. That's why I asked about the repeatibility in terms of microstructure in the as-built samples.
15. You already mentioned line 320-321 "The micrographs were segmented using Otsu’s 320 method [81, 82]." in the Methods section, so its kind of a repeat.
16. Could you please put error bar in x-axis (Mn composition from 9 EDS test) on Figure 7? Basically error bar of Mn composition based on the measurement.

I hope these remarks are useful to improve your manuscript. Thank you.

Reviewer 4 Report

An excellent manuscript, one of the first that this reviewer wiich describes phenomenological theory of so called "digital alloys".  Article is important step understanding  how the mechanical properties and structures changes in such materials.  

This reviewer believe that this article will become a highly cited piece of literature on metal additive manufacturing technology, and highly recommends it publications. 

One suggestions for future work (not necessary for this manuscript) is to map the  spatial distribution of mechanical properties properties of digital alloys by using phase-imaging AFM rapping mode technique, wherein the phase shift is corresponding to the modulus of elasticity. 

This reviewer recommends publications of article with the minor revision. The authors needs to describe in materials and methods in more details settings and additive manufacturing of the part using LENS process. Please specify type and model of the OPTOMEC machine you use ?  Please specify the settings.  Please insert a paragraph in materials and methods section. 

Round 2

Reviewer 3 Report

Dear Authors,

Thank you for revising properly the manuscript.

Good luck